# Appropriate-for-gestational-age infants who exhibit reduced antenatal growth velocity display postnatal catch-up growth

Emma J. McLaughlin[1]*, Richard J. Hiscock[1], Alice J. Robinson[2], Lisa Hui[1,2], Stephen Tong[1,2], Kirsten M. Dane[2], Anna L. Middleton[2], Susan P. Walker[1,2], Teresa M. MacDonald[1,2]

1 Department of Obstetrics and Gynaecology, University of Melbourne, Melbourne, Australia, 2 Mercy Perinatal, Mercy Hospital for Women, Melbourne, Australia

☯ These authors contributed equally to this work.
* emmamclaughlin111@gmail.com

## Abstract

### Background

Postnatally, small-for-gestational-age (SGA; birthweight <10th centile) infants who are growth restricted due to uteroplacental insufficiency (UPI) demonstrate 'catch-up growth' to meet their genetically-predetermined size. Infants who demonstrate slowing growth during pregnancy are those that cross estimated fetal weight centiles at serial ultrasound examinations. These infants that slow in growth but are born appropriate-for-gestational-age (AGA; ≥10th centile), exhibit antenatal, intrapartum and postnatal indicators of UPI. Here, we examine if and when these infants (labelled as AGA-FGR) also demonstrate catch-up growth like SGA infants, when compared with AGA infants with normal antenatal growth velocity (AGA-NG).

### Methods

We followed-up the infants of women who had previously undergone ultrasound assessment of fetal size at 28- and 36-weeks' gestation, enabling calculation of antenatal growth velocity. To assess postnatal growth, we asked parents to send their infant's growth measurements, up to two years post-birth, which are routinely collected through the state-wide Maternal-Child Health service. Infants with medical conditions affecting postnatal growth were excluded from the analysis. From the measurements obtained we calculated age-adjusted z-scores for postnatal weight, length and body mass index (BMI; weight(kg)/height (m$^2$)) at birth and 4, 8, 12, 18 and 24 months. We used linear spline regression modelling to predict mean weight, length and BMI z-scores at intervals post birth. Predicted mean age-adjusted z-scores were then compared between three groups; SGA, AGA with low antenatal growth (AGA-FGR; loss of >20 customised estimated fetal weight centiles), and AGA-NG to determine if catch-up growth occurred. In addition, we compared the rates of catch-up growth (defined as an increase in weight age-adjusted z-score of ≥0.67 over 1 year) between the groups with Fisher's exact tests.

**Data Availability Statement:** All relevant data are within the manuscript and its Supporting Information files.

**Funding:** Funding NHMRC Grant #1065854 to SW; Australian Government Research Training Program Scholarship to TM; Funding sources had no involvement in study design, collection or analysis of data, or in the writing or submission of this manuscript.

**Competing interests:** The authors report no conflict of interest.

## Results

Of 158 (46%) infant growth records received, 146 were AGA, with low antenatal growth velocity occurring in 34/146 (23.2%). Rates of gestational diabetes and SGA birthweight were higher in those lost to follow-up. Compared to AGA-NG infants, AGA-FGR infants had significantly lower predicted mean weight (p<0.001), length (p = 0.04) and BMI (p = 0.001) z-scores at birth. These significant differences were no longer evident at 4 months, suggesting that catch-up growth had occurred. As expected, the catch-up growth that occurred among the AGA-FGR was not as great in magnitude as that demonstrated by the SGA. When assessed categorically, there was no significant difference between the rate of catch-up growth among the AGA-FGR and the SGA. Catch-up growth was significantly more frequent among both the AGA-FGR and the SGA groups compared to the AGA-NG.

## Conclusions

AGA infants that have exhibited reduced antenatal fetal growth velocity also exhibit significant catch-up growth in the first 12 months of life. This finding represents further evidence that AGA fetuses that slow in growth during pregnancy do so due to UPI.

## Introduction

Uteroplacental insufficiency (UPI) is the most common cause of fetal growth restriction (FGR) [1]–a fetus that fails to achieve its genetically-predetermined growth potential [2]. Decreased nutrient and oxygen supply consequent to UPI results in failing growth, commonly manifesting as a low birthweight centile. As such, small-for-gestational-age (SGA; estimated fetal weight (EFW), or birthweight <10th centile) is the most commonly used proxy for FGR, as although not all SGA fetuses are growth restricted (some are constitutionally small), it does represent an at-risk cohort.

Being SGA is the greatest risk factor for stillbirth [3] but its legacy also extends into postnatal life among survivors. After removal from their suboptimal intrauterine environment, growth restricted infants demonstrate catch-up growth [4–6]. In the first 6 months of life their growth rate accelerates such that they return to near their genetically-predetermined height and weight [7, 8]. Catch-up growth therefore represents postnatal evidence of UPI.

Postnatal catch-up growth is both beneficial and detrimental. While it provides early survival and neurodevelopmental advantages [9, 10], there is evidence that the development of cardiometabolic disease in adulthood is related to early key growth periods. These include the period of *in utero* fetal growth resulting in low birth weight, and early infancy where catch-up growth occurs [11, 12].

The Developmental Origins of Health and Disease hypothesis [13, 14] proposes that the relationship between FGR and adult disease is due to fetal programming *in utero*. While vascular redistribution to critical circulations in a poor nutritional environment increases fetal survival, this carries a long-term legacy with reduced nephron, pancreatic cell and cardiomyocyte endowment. An alternate explanation is the 'catch-up growth hypothesis' where the adiposity rapidly gained by some SGA infants in the early postnatal period initiates a cascade of metabolic risk factors [15]. These alternate perspectives highlight the important implications of both fetal and infant growth trajectory on adult health.

Notably, 50% of stillbirths occur in fetuses who are not small but are classified as 'appropriate-for-gestational-age' (AGA; ≥10th centile) [3]. Our recent research in the Fetal Longitudinal Assessment of Growth (FLAG) Study revealed that even AGA fetuses who demonstrate slowing of growth trajectory across the third trimester, exhibit signs of UPI during the antenatal, intrapartum and postpartum periods [16]. This suggests that decreased antenatal growth velocity may be an important, previously unrecognised, measure of UPI, and stillbirth risk. We therefore further propose that these AGA infants may be at risk of similar adverse health outcomes as SGA infants.

In this childhood follow-up to the FLAG study we aimed to: i) determine whether AGA infants with slowing antenatal growth demonstrate catch-up growth; ii) compare AGA catch-up growth to that of the SGA infants; and iii) determine in which postnatal time period catch-up growth occurs.

## Materials and methods

### FLAG study overview

The FLAG study was a prospective longitudinal cohort study conducted in 2015 and 2016 at the Mercy Hospital for Women, a tertiary maternity hospital in Melbourne. Its protocol and findings have previously been published [16]. Women underwent two ultrasound examinations, at 28- and 36-weeks' gestation. The change in EFW centile over exactly eight weeks was calculated and termed the "EFW growth velocity". The cohort consisted of three groups: those born SGA; those born AGA with no antenatal slowing of growth; and those born AGA with reduced antenatal EFW growth velocity defined as a fall in EFW of >30 centiles over exactly eight weeks. The last two groups were compared to demonstrate that AGA fetuses with reduced growth velocity showed evidence of UPI in the antenatal, intrapartum and postpartum periods [16].

### FLAG follow-up study overview

Between February and May 2018, we invited the 347 FLAG participants to complete a self-administered postnatal survey (S1 Appendix). This collected data regarding infant medical admissions and treatments during the first two years of life; and the infant weight, length and head circumference measurements. These measurements are routinely performed during maternal-child-health visits at 1, 2, 4 and 8 weeks; 4, 8, 12 and 18 months; and at 2 years of age and recorded in the child's analogue 'My Health, Learning & Development Record' [17]. 80–90% of SGA infants will have 'caught-up' in their postnatal growth by two years of age [4–6, 18–26] and so this was the pre-determined end-point for our analyses.

### Ethics statement

The FLAG study was approved by the Mercy Health Research Ethics Committee, Ethics Approval Number R14/12, and amendments to include this follow-up study were approved in February 2018. Written informed consent was obtained from all participants.

### Growth assessment

**Fetal growth assessment.** The Gestation Related Optimal Weight (GROW) software [27] was used in the FLAG study to generate customised EFW and birthweight centiles. The current study focussed on comparing antenatal and postnatal growth trajectory, not predictive performance for adverse outcome for which the original study was powered. Given the different outcome being interrogated, and expected loss to follow up, we used a more sensitive cut-

off for antenatal growth velocity than in the original FLAG study to ensure a sufficient number of cases of low antenatal growth velocity. Fetuses who exhibited an antenatal decrease in EFW of >20 customised centiles over eight weeks were defined as experiencing low antenatal growth velocity and to potentially be growth restricted (labelled as AGA-FGR). The remainder of the AGA fetuses were defined as demonstrating normal antenatal growth (AGA-NG).

**Postnatal growth assessment.** For the current study, SGA was defined as birthweight <10th centile, and AGA as birthweight ≥10th centile according to the World Health Organisation (WHO) growth charts [28].

On the basis of fetal growth velocity and infant birthweight centile, the follow-up cohort was again divided into three groups: the SGA; the AGA with normal antenatal growth velocity (AGA-NG); and the AGA with low antenatal growth velocity (AGA-FGR).

**Catch-up growth.** We wished to evaluate the postnatal growth trajectory of AGA-FGR infants compared to the AGA-NG and SGA groups.

Using the WHO child growth standards software [29], we calculated standard deviation z-scores, adjusted for sex and age, for weight, length and body mass index (BMI; weight(kg)/height(m)$^2$). We then determined the mean weight, length and BMI z-scores at birth and at 4, 6, 8, 12, 18 and 24 months for each of the three groups. Using the mean z-scores derived from the raw data, we fit curves plotting the change in predicted mean z-scores over time and then compared the mean predicted z-scores between the 3 groups. If one group had significantly lower predicted mean weight, length and/or BMI z-scores than the AGA-NG group at one point, and then after a period of time there was no longer a significant difference seen, then catch-up growth was deemed to have occurred.

Secondly, we examined catch-up growth in weight as a dichotomous variable using the individual difference between weight-for-age Z-scores at birth and 12 months. Those that had a change in weight-for-age Z-score greater than or equal to +0.67 were defined as having shown catch-up growth as proposed by Ong et al. [30] The rate and relative risk of catch-up growth between the groups was compared. It is important to note however, that although widely used [31, 32], this definition of catch-up growth has not been formally validated.

## Statistical analysis

**Study participants.** Maternal characteristics and birth outcome data were compared between both: recruited follow-up study participants and the eligible women who did not participate, to check for selection bias; and cases of low antenatal velocity and the remainder of the AGA cohort, the subjects of our primary comparison. Hypothesis testing used the unpaired t-test (normally-distributed) or Mann–Whitney test (not normally-distributed) for continuous data, and Fisher's exact test for categorical data.

**Antenatal growth velocity and catch-up growth.** Growth curve modelling was performed using linear spline regression, to fit curves plotting the change in predicted mean weight, length and BMI z-scores over time, for each antenatal growth category. User determined knots were placed at five time points (4, 6, 12, 18 and 24 months). We also fitted growth curves using both: restricted cubic spline model with knots placed at the quartiles of the time distribution and an interaction between time and antenatal growth category; and locally weighted kernel regression. All statistical methods produced extremely similar results, reinforcing the validity of our linear spline regression approach (Data not shown, available upon request).

The relationships between EFW growth velocity and postnatal weight, length and BMI were assessed in two ways, without adjustment for any other variables. First predicted mean difference and associated 95% confidence limits for weight, height and BMI z-scores were

calculated at birth, 2, 4, 6, 12, 18 and 24 months for the three comparisons between antenatal growth categories (SGA & AGA-FGR, SGA & AGA-NG, and AGA-FGR & AGA-NG). We defined catch-up growth as occurring if the SGA or AGA-FGR group infants were significantly smaller than the AGA-NG group infants in weight, length and/or BMI at one time point, followed by no significant difference in infant size months later. Secondly, we assessed rates and relative risks of catch-up growth between the three groups. Catch-up growth was defined as an increase in weight age-adjusted z-score of $\geq 0.67$ between birth and 12 months of age [30]. Analysis used Fisher's exact test across the three antenatal growth categories. If the null hypothesis of no overall difference in proportions was rejected, between pair testing was performed. Statistical analyses were performed using GraphPad Prism software version 7.0d for Mac OS X [33] and the nonparametric series regression suite within Stata v16 [34]. Significance level was two-sided and set at 0.05. No adjustment for multiple comparisons for either significance testing or confidence interval width was performed.

## Results

### Study participants

Between February-May 2018, 158 (46%) of the 347 eligible women who completed the FLAG study were recruited. Of these, 2 infants (1.3%) were excluded due to medical illnesses that were deemed to have the potential to confound postnatal growth, leaving 156 (45%) infants for the final analysis (S1 Fig).

We compared the maternal characteristics and delivery outcomes of the 156 women we recruited to the 189 non-responders to assess for selection bias (S1 Table). When compared to non-responders, those who returned data were slightly older, more likely to have spontaneous onset of labour and less likely to have gestational diabetes or an SGA infant using customised weight centiles. Importantly there were no significant differences in gestational age at birth, or antenatal growth velocity between the two groups, indicating that those recruited were representative of the original FLAG cohort.

Out of the 156 eligible responders, 10 infants were SGA and 146 AGA. Given our primary question concerned whether AGA-FGR infants demonstrate catch-up growth compared to the AGA-NG, as a sign of UPI occurring among the AGA, we compared their maternal and pregnancy characteristics (Table 1). There were no significant differences between maternal characteristics and delivery outcomes for AGA infants with low antenatal growth velocity (AGA-FGR) compared to AGA infants with normal antenatal growth velocity (AGA-NG).

### Postnatal growth velocity according to antenatal growth velocity

We first used the raw infant measurement data and linear spline regression to predict the mean (and 95% confidence interval) weight, length and BMI z-scores for the three groups at birth, 4, 6, 12, 18 and 24 months (Figs 1 and 2). We then analysed the relationship between antenatal growth velocity and postnatal growth between the three groups by assessing whether the predicted mean weight, length and BMI z-scores were significantly different between the three groups at birth, 2, 4, 6, 12, 18 and 24 months (Fig 2 and Table 2). At birth there were statistically significant differences in predicted mean weight (p<0.001), length (p = 0.04) and BMI (p = 0.001) z-scores between the AGA-FGR and AGA-NG infants. The differences were still significantly different at 2 months for all 3 growth parameters. By 4 months of age and beyond there were no longer significant differences in mean predicted weight, length or BMI z-scores, confirming catch-up growth had occurred; and indicating the majority of catch-up growth occurred between birth and four months of age.

**Table 1. Maternal characteristics and delivery outcomes of recruited appropriate-for-gestational-age participants overall and comparison between infants with low- and normal-antenatal growth velocity.**

| | Total AGA | AGA-FGR | AGA-NG | P |
|---|---|---|---|---|
| | (n = 146) | (n = 34) | (n = 112) | |
| Age (years) | 31.6 (3.7) | 31.2 (3.4) | 31.7 (3.9) | 0.5 |
| Booking BMI (kg/m$^2$) | 23.6 [21.3–26.5] | 23.9 [21.2–26.7] | 23.4 [21.3–26.4] | 0.5 |
| Smoking status | | | | 0.7 |
| Current smoker | 2 (1%) | 0 (0%) | 2 (2%) | |
| Ex-smoker | 39 (27%) | 8 (24%) | 31 (28%) | |
| Never | 104 (71%) | 26 (77%) | 78 (70%) | |
| No information | 1 (1%) | 0 (0%) | 1 (1%) | |
| Gestational hypertension or pre-eclampsia | 18 (12%) | 7 (21%) | 11 (10%) | 0.1 |
| Gestational diabetes mellitus | 10 (7%) | 3 (9%) | 7 (6%) | 0.7 |
| Onset of labour | | | | 0.5 |
| Spontaneous labour | 78 (53%) | 21 (62%) | 57 (51%) | |
| Induction of labour | 61 (42%) | 12 (35%) | 49 (44%) | |
| No labour | 7 (5%) | 1 (3%) | 6 (5%) | |
| Mode of delivery | | | | 0.08 |
| Normal vaginal delivery | 62 (43%) | 21 (62%) | 41 (37%) | |
| Instrumental delivery | 50 (34%) | 8 (24%) | 42 (38%) | |
| Emergency caesarean | 28 (19%) | 4 (12%) | 24 (21%) | |
| Elective caesarean | 6 (4%) | 1 (3%) | 5 (5%) | |
| Gestational age at delivery (weeks) | 39.5 [38.9–40.5] | 39.7 [39.1–40.3] | 39.4 [38.7–40.6] | 0.8 |
| Infant Sex (M:F) | 80:66 | 21:13 | 59:53 | 0.4 |

Data presented as mean (standard deviation) or median [interquartile range] depending on distribution for continuous variables, and as number (%) for categorical variables. Note: some columns do not total 100% on account of rounding to nearest whole number.

SGA small-for-gestational-age; AGA appropriate-for-gestational-age; FGR fetal growth restriction; NG normal antenatal growth; BMI body mass index; M male; F female.

Our suspicions that the AGA-FGR group represents a less severe form of UPI than being born SGA were also confirmed. SGA infants had significantly lower predicted mean weight, length and BMI z-scores not just compared to AGA-NG infants at birth and 2 months, but also when compared to the AGA-FGR cohort (Table 2). Figs 1 and 2 show that the SGA group demonstrated more rapid catch-up growth than the AGA-FGR, and that the greatest catch-up growth occurred in the first 4 months for the SGA cohort also. In fact, the shape of the growth curves in Figs 1 and 2 show that the AGA-FGR group demonstrated a similar, physiological fall in z-scores like the AGA-NG group, but of a lesser magnitude–leading to their catch-up growth. In contrast, the SGA cohort's catch-up growth took the form of a steady increase in z-scores without the physiological fall seen among the AGA. Despite the catch-up growth that occurred, SGA infants were still significantly smaller than AGA-NG infants at 6 months for predicted mean weight (p = 0.01) and height (p = 0.006) z-scores; and at 12 months for predicted mean weight (p = 0.004) and BMI (p = 0.04) z-scores. The SGA group also had a significantly lower predicted mean weight (p = 0.03) z-score at 12 months compared to the AGA-FGR group.

## Catch-up growth according to antenatal growth velocity and birthweight

We calculated the change in weight-for-age Z-score between birth and 12 months of age if there was a weight recorded for an infant within one month of their first birthday. This data

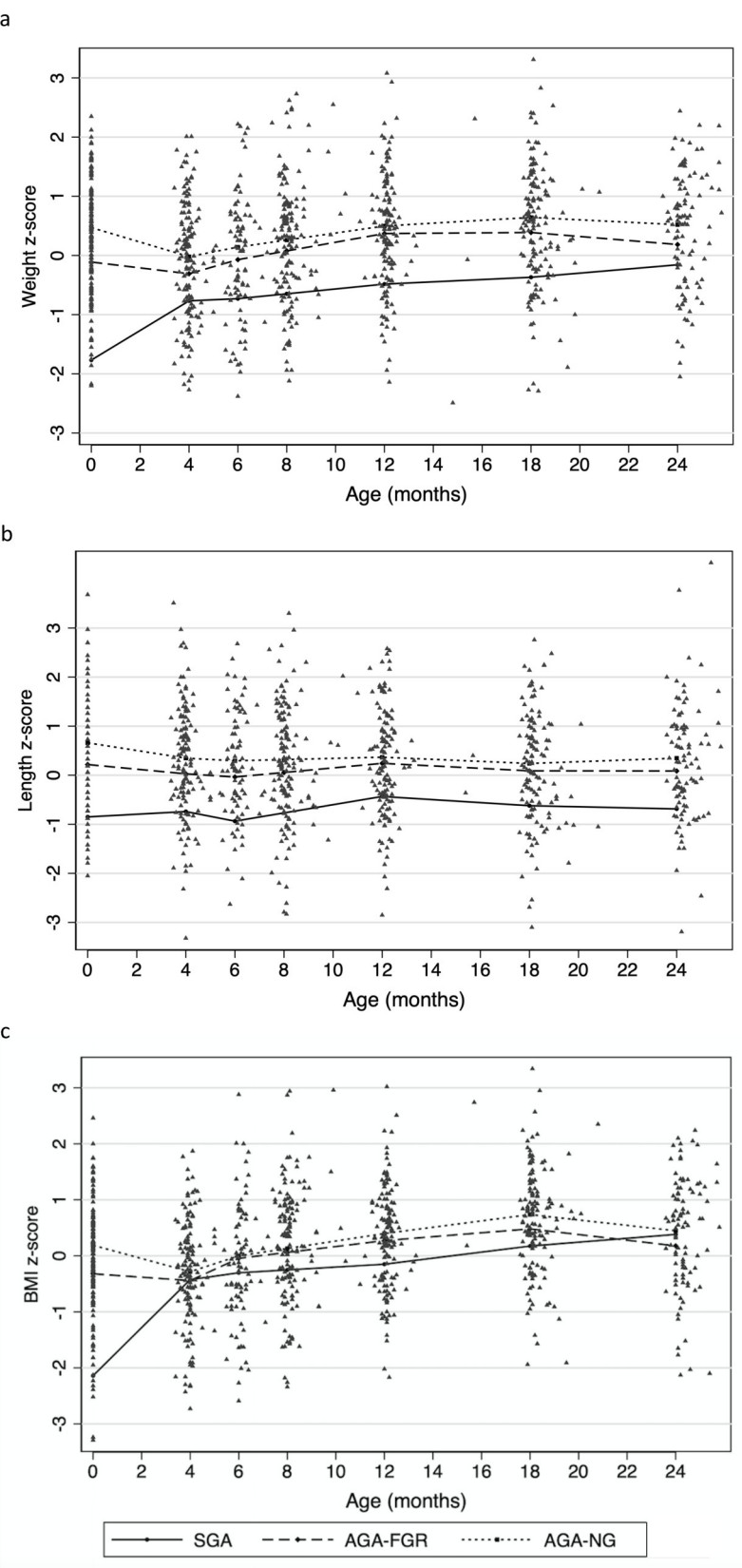

**Fig 1.** Infant (a) weight, (b) length, and (c) BMI z-scores converted to predicted mean z-scores and growth curves according to antenatal group. Each triangle represents an individual's z-score at one of seven (birth or 4, 6, 8, 12, 18 or 24 months) timepoints at which measurements were performed. Lines represent the predicted mean growth for each of the 3 antenatal groups as plotted by linear spline regression. BMI body mass index; SGA small-for-gestational-age; AGA appropriate-for-gestational-age; FGR fetal growth restriction; NG normal antenatal growth.

was available for 139 (95%) of our respondents. When catch-up growth was assessed as a dichotomous outcome (change in weight-for-age Z-score $\geq +0.67$) between birth and 12 months of age between all three groups, a significant difference was found (p = 0.004). When between pair testing was then performed, the SGA and AGA-FGR cohorts were both found to have a higher probability of catch-up growth when compared to the AGA-NG group (Table 3). For the SGA cohort the probability was approximately tripled, and for the AGA-FGR cohort, it was doubled. The relative risks (RR) of catch-up growth for the SGA and the AGA-FGR infants compared to the AGA-NG group between birth and 12 months were 2.8 (p = 0.01) and 2.0 (p = 0.01) respectively. The probability of catch-up growth was not significantly increased for the SGA group compared to the AGA-FGR.

## Discussion

### Main findings

In this study we have tracked growth trajectory from the third trimester of pregnancy until two years of postnatal age. We have demonstrated that catch-up growth–hypothesised to be associated with some long-term health decrements—occurs more frequently in AGA fetuses who have demonstrated slowed antenatal growth, than those who maintained growth in the third trimester. This represents further evidence of UPI occurring among fetuses with reduced antenatal growth velocity, even if born AGA. It suggests that these infants may represent an unrecognised cohort at risk of not only the antenatal, intrapartum and neonatal consequences of UPI, but the infant and adult health decrements associated with FGR.

### Strengths and limitations

FLAG was a prospective study evaluating the longitudinal assessment of fetal growth. In FLAG we showed important associations between the AGA-FGR cohort and antenatal, intrapartum and neonatal features of uteroplacental insufficiency; fetal cerebral redistribution at 36 weeks, umbilical artery acidosis following the hypoxic challenge of labour, and reduced neonatal body fat [16]. The aim of this follow-up study was to determine whether these short-term measures of UPI are subsequently followed by catch-up growth in infancy with the attendant risk of longer-term UPI-related health-decrements. Further strengths of this follow-up study include the evaluation for selection bias where we confirmed the participants were representative of the original FLAG cohort. We also had a reasonably good response rate (46%) for a questionnaire administered to a cohort who had delivered their infants at least two years earlier. Finally, we ensured a robust statistical comparison for growth across time, by using gestation specific weight centiles and postnatal age specific z-scores to facilitate meaningful comparison and improve the precision of assessment.

The limitations of this study were that we were unable to obtain measurements at every timepoint for every child, and that each child's measurements were performed by different clinicians in the community. Although the maternal-child-health nurses are trained to measure and enter growth information using WHO growth standards, there remains a possibility of inter- and intra-observer variability which we could not account for. However, this does represent a real-life reflection of postnatal growth assessment which increases the generalisability of

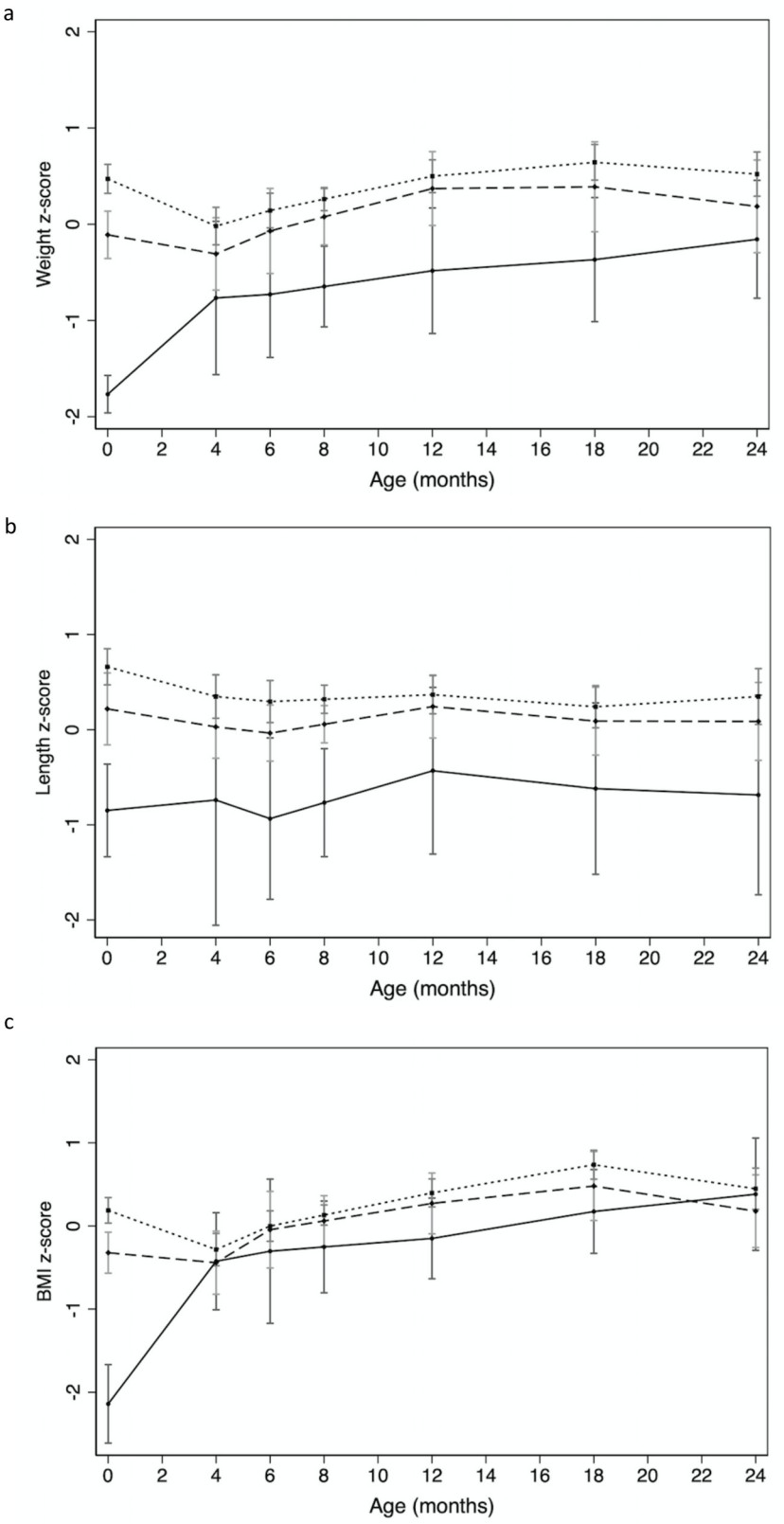

**Fig 2.** Comparison of predicted mean (a) weight, (b) length, and (c) BMI z-scores between antenatal groups. Data presented as predicted mean and 95% confidence intervals for each of the three groups at birth, and 4, 6, 8, 12, 18 and 24 months. Lines connecting the predicted mean z-scores for each group represent predicted mean postnatal growth trajectory as modelled by linear spline regression. BMI body mass index; SGA small-for-gestational-age; AGA appropriate-for-gestational-age; FGR fetal growth restriction; NG normal antenatal growth.

our findings. Overall, our cohort was relatively small, as such, our results require validation in a larger study. Finally, this study interrogating fetal and postnatal growth rates is based upon the developmental origins of health and disease hypothesis. While this is a theory to which we ascribe, it is not universally accepted, with some research groups considering that the relationships between fetal growth and adult disease seen may be due to over-adjustments because of the analysis methods used [35–38]. In addition, the limitations of the original FLAG study have already been discussed [16], but relevant here is that estimated fetal weight as measured by ultrasound is only accurate to within 10% of the true weight 80% of the time [39].

## Interpretation of the findings and comparison with other studies

Only two other studies have examined the relationship between fetal growth velocity and catch-up growth in AGA infants—the CASyMIR cohort [40] and the Generation R Study [41, 42]. In the CASyMIR cohort, 94 infants at increased risk of FGR had serial third trimester ultrasound examinations and were subsequently followed-up with BMI, skinfold thickness and hormonal concentration measurements at 4, 9 and 12 months of age. Low antenatal growth rate was defined as >25 centile loss between 22 weeks' gestation and birth, and their results were in agreement with ours. At birth, they found that infants with low antenatal growth velocity were shorter, lighter and had lower BMIs. At birth they also had higher insulin sensitivity and lower leptin levels and body fat percentage. By 4 months of age there was no significant difference in BMI, body fat percentage, weight or length–similar to the catch-up growth by 4 months of age that we observed [40].

The Generation R Study, a prospective cohort study of 7959 infants, also examined the relationship between low fetal growth velocity, delivery outcomes, infant catch-up growth and childhood cardiovascular outcomes [42]. Similar to the CASyMIR cohort, growth velocity was defined between a second trimester EFW and birthweight; not between serial antenatal ultrasounds as was done in FLAG. Generation R also found that decreased antenatal growth velocity (defined as a loss of >40 weight centiles) was associated with catch-up growth at two years of age.

An important difference from our study is that the analysis of catch-up growth in both the CASyMIR cohort and the Generation R Study did not consistently distinguish between SGA and AGA with low antenatal growth velocity. Although these analyses were reportedly conducted by Broere-Brown et al. [42] and yielded the same results, they were not published. This means that those showing catch-up growth may have been dominated by those born SGA. Furthermore, the Generation R Study did not differentiate between preterm and term gestations in their AGA cohort [42]. While their findings support our results, we know that both SGA and preterm AGA infants are more likely to demonstrate catch-up growth [22, 43]. Our study provides important evidence that slowing growth among term AGA infants identifies a currently under-recognised cohort that may be at increased risk of both short and long-term consequences of placental insufficiency.

It has been proposed that the long-term cardiovascular risks associated with catch-up growth are, in part, due to an accumulation of adiposity preferential to lean mass—deemed 'catch-up fat' [44]. While catch-up growth was more evident in weight than length between birth and 4 months in our cohort, at no point in the first 2 years was BMI greater in AGA-FGR

**Table 2. Comparison of predicted mean growth parameter z-scores between SGA, AGA-FGR and AGA-NG infants.**

| Growth parameter and timepoint | Difference (SGA vs AGA-FGR) | | Difference (SGA vs AGA-NG) | | Difference (AGA-FGR vs AGA-NG) | |
|---|---|---|---|---|---|---|
| | Predicted mean (95% CI) | *P*\* | Predicted mean (95% CI) | *P*\* | Predicted mean (95% CI) | *P*\* |
| **Weight z-score** | | | | | | |
| Birth | 1.656 | < 0.001 | 2.234 | < 0.001 | 0.582 | < 0.001 |
| | (1.342 to 1.970) | | (1.991 to 2.484) | | (0.293 to 0.870) | |
| 2 months | 1.057 | < 0.001 | 1.493 | < 0.001 | 0.436 | 0.001 |
| | (0.590 to 1.524) | | (1.064 to 1.921) | | (0.182 to 0.690) | |
| 4 months | 0.458 | 0.31 | 0.748 | 0.08 | 0.290 | 0.18 |
| | (-0.425 to 1.341) | | (-0.074 to 1.569) | | (-0.134 to 0.714) | |
| 6 months | 0.659 | 0.10 | 0.872 | 0.01 | 0.212 | 0.38 |
| | (-0.132 to 1.451) | | (0.191 to 1.552) | | (-0.266 to 0.690) | |
| 12 months | 0.855 | 0.03 | 0.982 | 0.004 | 0.128 | 0.55 |
| | (0.096 to 1.613) | | (0.307 to 1.658) | | (-0.292 to 0.548) | |
| 18 months | 0.757 | 0.06 | 1.011 | 0.003 | 0.254 | 0.32 |
| | (-0.040 to 1.553) | | (0.340 to 1.682) | | (-0.247 to 0.757) | |
| 24 months | 0.342 | 0.39 | 0.678 | 0.04 | 0.335 | 0.22 |
| | (-0.437 to 1.121) | | (0.023 to 1.332) | | (-0.198 to 0.869) | |
| **Length z-score** | | | | | | |
| Birth | 1.067 | 0.001 | 1.509 | < 0.001 | 0.442 | 0.04 |
| | (0.450 to 1.685) | | (0.985 to 2.033) | | (0.019 to 0.866) | |
| 2 months | 0.918 | 0.02 | 1.299 | < 0.001 | 0.381 | 0.009 |
| | (0.174 to 1.662) | | (0.580 to 2.017) | | (0.095 to 0.667) | |
| 4 months | 0.769 | 0.27 | 1.088 | 0.11 | 0.319 | 0.12 |
| | (-0.591 to 2.128) | | (-0.251 to 2.426) | | (-0.1083 to 0.721) | |
| 6 months | 0.899 | 0.05 | 1.230 | 0.006 | 0.331 | 0.08 |
| | (-0.001 to 1.80) | | (0.352 to 2.108) | | (-0.038 to 0.701) | |
| 12 months | 0.675 | 0.16 | 0.799 | 0.08 | 0.124 | 0.53 |
| | (-0.263 to 1.613) | | (-0.101 to 1.700) | | (-0.263 to 0.512) | |
| 18 months | 0.698 | 0.16 | 0.860 | 0.07 | 0.162 | 0.59 |
| | (-0.274 to 1.669) | | (-0.071 to 1.790) | | (-0.258 to 0.582) | |
| 24 months | 1.062 | 0.11 | 1.086 | 0.09 | 0.025 | 0.93 |
| | (-0.240 to 2.363) | | (-0.171 to 2.344) | | (-0.497 to 0.546) | |
| **BMI z-score** | | | | | | |
| Birth | 1.818 | < 0.001 | 2.327 | < 0.001 | 0.509 | 0.001 |
| | (1.284 to 2.351) | | (1.829 to 2.824) | | (0.218 to 0.800) | |
| 2 months | 0.990 | < 0.001 | 1.234 | < 0.001 | 0.338 | 0.01 |
| | (0.462 to 1.338) | | (0.839 to 1.629( | | (0.078 to 0.590) | |
| 4 months | -0.018 | 0.96 | 0.141 | 0.65 | 0.159 | 0.47 |
| | (-0.714 to 0.679) | | (-0.475 to 0.757) | | (-0.268 to 0.585) | |
| 6 months | 0.258 | 0.61 | 0.302 | 0.51 | 0.043 | 0.86 |
| | (-0.725 to 1.242) | | (-0.586 to 1.189) | | (-0.453 to 0.540) | |
| 12 months | 0.421 | 0.18 | 0.547 | 0.04 | 0.126 | 0.54 |
| | (-0.187 to 1.030) | | (0.033 to 1.062) | | (-0.277 to 0.529) | |
| 18 months | 0.309 | 0.36 | 0.562 | 0.04 | 0.253 | 0.27 |
| | (-0.345 to 0.962) | | (0.027 to 1.096) | | (-0.196 to 0.703) | |

(*Continued*)

**Table 2.** (Continued)

| Growth parameter and timepoint | Difference (SGA vs AGA-FGR) | | Difference (SGA vs AGA-NG) | | Difference (AGA-FGR vs AGA-NG) | |
|---|---|---|---|---|---|---|
| | Predicted mean (95% CI) | *P** | Predicted mean (95% CI) | *P** | Predicted mean (95% CI) | *P** |
| 24 months | -0.256 | 0.59 | 0.081 | 0.85 | 0.336 | 0.24 |
| | (-1.175 to 0.665)) | | (-0.734 to 0.896) | | (-0.228 to 0.901) | |

* based upon z test, unadjusted for multiple comparisons; SGA small-for-gestational-age; AGA appropriate-for-gestational-age; FGR fetal growth restriction; NG normal antenatal growth; CI confidence interval; BMI body mass index.

than in AGA-NG infants. This suggests that these infants have not caught up disproportionately in weight. However, as we did not assess body fat percentage or other more sophisticated body composition measures postnatally, we cannot rule out the possibility that this catch-up growth was due to an excess accumulation of fat mass.

It is emerging that early catch-up growth (i.e. in the first two years of life) may be particularly obesogenic. Rotevatn et al. [45], Stettler et al. [46] and Monteiro et al. [47] found that rapid weight gain in the first year of life was associated with being overweight at 2, 7, and 14–16 years of age, respectively, independent of socioeconomic status. This is supported by a recent meta-analysis which found that the odds of being overweight from childhood to adulthood was 3.66 times greater in infants who experienced catch-up growth [48].

Ibanez et al. [49] found that although there was no difference in height, weight and BMI between SGA and AGA children at 2, 3 or 4 years of age, by 4 years of age SGA infants had significantly greater body fat percentage. As childhood body fat percentage is a predictor of adult obesity [50], this may indicate that early catch-up growth confers long-term metabolic risk. While we saw no difference in BMI at 2 years of age between AGA-FGR and AGA-NG infants, the fact that AGA-FGR infants demonstrated catch-up growth in the first year of life raises the possibility that this is an unrecognised group at increased risk of cardiovascular disease.

## Clinical and research implications

Our data adds to the evidence that AGA infants who are observed by antenatal ultrasound to slow in growth are experiencing UPI and are therefore growth-restricted., even if they are not born SGA. AGA-FGR infants may represent an intermediary between SGA and AGA-NG as they undergo significant, although lesser, catch-up growth than SGA infants. These infants may therefore be at increased risk of the antenatal, intrapartum, postnatal, childhood and adulthood complications that FGR and catch-up growth confer compared to AGA-NG infants.

Serial assessment of fetal growth may have a role in the clinical management of pregnancy, but with considerable cost and access implications. For women already undergoing ultrasound

**Table 3. Relative risk of catch-up growth in small-for-gestational-age infants compared to appropriate-for-gestational-age infants.**

| Cohort 1 (n) | n (%) with catch-up growth | Cohort 2 (n) | n (%) with catch-up growth | RR (95% CI) of catch-up growth if Cohort 1 | *P** |
|---|---|---|---|---|---|
| SGA (n = 9) | 6 (66.7%) | AGA-FGR (n = 31) | 15 (48.4%) | 1.4 (0.7–2.3) | 0.46 |
| SGA (n = 9) | 6 (66.7%) | AGA-NG (n = 99) | 24 (24.2%) | 2.8 (1.4–4.5) | 0.01 |
| AGA-FGR (n = 31) | 15 (48.4%) | AGA-NG (n = 99) | 24 (24.2%) | 2.0 (1.2–3.2) | 0.01 |

* based upon Fisher exact test, unadjusted for multiple comparisons; catch-up growth (increase in weight age-adjusted z-score of 0.67 or more between birth and 12 months of age); RR relative risk; CI confidence interval; SGA small-for-gestational-age; AGA appropriate-for-gestational-age; FGR fetal growth restriction; NG normal antenatal growth.

assessment due to risk factors, AGA infants with slowing growth may warrant more intensive surveillance as would be afforded to SGA infants. This study suggests this surveillance should potentially be extended into childhood to minimise the long-term health decrements associated with placental insufficiency and growth restriction.

## Conclusions

AGA infants who experience low growth velocity in the third trimester of pregnancy exhibit catch-up growth in both length and weight in the first four months of life. This study adds to our previous research which showed that these infants demonstrate features suggestive of uteroplacental insufficiency that are typically associated with SGA fetuses. Our findings suggest that these infants may also be at increased risk of developing the long-term health decrements traditionally associated with infants born SGA.

## Supporting information

**S1 Fig. Study profile.**
(TIF)

**S1 Table. Demographic and delivery characteristics of responding participants compared to eligible women who did not respond.**
(DOCX)

**S1 Appendix. Patient questionnaire.**
(PDF)

**S1 Dataset. FLAG follow-up dataset.**
(XLSX)

**S1 File.**
(ZIP)

## Acknowledgments

We wish to thank all staff members at the Mercy Hospital for Women and the University of Melbourne for their assistance in conducting this study.

## Author Contributions

**Conceptualization:** Lisa Hui, Stephen Tong, Susan P. Walker, Teresa M. MacDonald.

**Data curation:** Emma J. McLaughlin, Teresa M. MacDonald.

**Formal analysis:** Emma J. McLaughlin, Richard J. Hiscock, Teresa M. MacDonald.

**Funding acquisition:** Susan P. Walker, Teresa M. MacDonald.

**Investigation:** Emma J. McLaughlin, Alice J. Robinson, Kirsten M. Dane, Anna L. Middleton, Teresa M. MacDonald.

**Methodology:** Emma J. McLaughlin, Susan P. Walker, Teresa M. MacDonald.

**Project administration:** Stephen Tong, Susan P. Walker, Teresa M. MacDonald.

**Supervision:** Lisa Hui, Stephen Tong, Susan P. Walker, Teresa M. MacDonald.

**Visualization:** Richard J. Hiscock.

**Writing – original draft:** Emma J. McLaughlin.

**Writing – review & editing:** Emma J. McLaughlin, Richard J. Hiscock, Lisa Hui, Stephen Tong, Susan P. Walker, Teresa M. MacDonald.

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
