## [Decision Letter · Decision Letter 0]

29 Jan 2020

PONE-D-19-31762

Appropriate-for-gestational-age infants who exhibit reduced antenatal growth velocity display postnatal catch-up growth

PLOS ONE

Dear Dr McLaughlin,

Thank you for submitting your manuscript to PLOS ONE. After careful consideration, we feel that it has merit but does not fully meet PLOS ONE’s publication criteria as it currently stands. Therefore, we invite you to submit a revised version of the manuscript that addresses the points raised during the review process.

Would you choose to submit a revised version of your manuscript, please address all the comments made by the reviewers.

We would appreciate receiving your revised manuscript by Mar 14 2020 11:59PM. To enhance the reproducibility of your results, we recommend that if applicable you deposit your laboratory protocols in protocols.io, where a protocol can be assigned its own identifier (DOI) such that it can be cited independently in the future. For instructions see: http://journals.plos.org/plosone/s/submission-guidelines#loc-laboratory-protocols

We look forward to receiving your revised manuscript.

Kind regards,

Umberto Simeoni

Academic Editor

PLOS ONE

Journal Requirements:

2. Please amend either the title on the online submission form (via Edit Submission) or the title in the manuscript so that they are identical.

3. Your ethics statement must appear in the Methods section of your manuscript. If your ethics statement is written in any section besides the Methods, please move it to the Methods section and delete it from any other section. Please also ensure that your ethics statement is included in your manuscript, as the ethics section of your online submission will not be published alongside your manuscript.

Reviewers' comments:

Reviewer's Responses to Questions

**Comments to the Author**

1. Is the manuscript technically sound, and do the data support the conclusions?

Reviewer #1: Partly

Reviewer #2: Partly

2. Has the statistical analysis been performed appropriately and rigorously? 

Reviewer #1: I Don't Know

Reviewer #2: No

3. Have the authors made all data underlying the findings in their manuscript fully available?

Reviewer #1: Yes

Reviewer #2: Yes

4. Is the manuscript presented in an intelligible fashion and written in standard English?

Reviewer #1: Yes

Reviewer #2: Yes

5. Review Comments to the Author

Reviewer #1: While the Flag study used a more informed measure of EFW (used the slowing of growth) however this is not identified in the abstract and paper introduction and the abstract introduction only refers to weight <10th percentile.

The abstract needs to mention the high loss to follow which was ~ 50%, and the high proportion of GDM in those lost to follow up compared to respondents.

The word “responses” was used in the abstract but it is not clear what it means. The abstract needs to mention that the postnatal anthropometric measurements were obtained by parent report.

The study makes the point that not all infants with a birthweight >10th percentile have avoided growth restriction. However, the counter problems exists as well, since the 10th percentile is an arbitrary statistical cut-off. Not all infants that are born SGA with a birthweight <10th percentile have utero placental insufficiency. The paper’s introduction should mention that some infants in the SGA category are small normal infants.

In contrast to the statement “strong evidence that the development of cardiometabolic disease in adulthood is related to early key growth periods”, some of these presumed effects may be seen due to the analysis methods used. It would be valuable to consider possible over adjustment as described in the following references:

1. Kramer MS, Zhang X, Dahhou M, Yang S, Martin RM, Oken E, et al. Does fetal growth restriction cause later obesity? Pitfalls in analyzing causal mediators as confounders. Am J Epidemiol. 2017 Apr;185(7):585–90.

2. Huxley R, Neil A, Collins R. Unravelling the fetal origins hypothesis: is there really an inverse association between birthweight and subsequent blood pressure? Lancet 2002 Aug;360(9334):659–65.

3. Paneth N, Ahmed F, Stein AD. Early nutritional origins of hypertension: a hypothesis still lacking support. J Hypertens Suppl. 1996 Dec;14(5):S121–9.

4. Tu Y-K, West R, Ellison GTH, Gilthorpe MS. Why evidence for the fetal origins of adult disease might be a statistical artifact: the “reversal paradox” for the relation between birth weight and blood pressure in later life. Am J Epidemiol. 2005;161:27–32.

It should be noted that there is a poor association between estimated foetal weight and birthweight.

This study used percentiles to evaluate antenatal growth, but percentiles become meaningless at the extremes. SD scores, which were used in some analyses, are better than percentiles as they are meaningful over the full range of the values.

What precision was used to calculate the standard deviation scores, was it the WHO monthly data or daily data?

Analysis: were any variables adjusted for in the regression analysis? It is not necessary to adjust, in fact it is appropriate to report crude analyses, it just needs to be clear if adjustments were made in any of the analyses/

Line 188: “Fishers exact test was used was used to ascertain the relative risk of catch up growth” does not make sense since it is a statistical test.

Results

While there were significant differences in birth size between the FGR & NG infants, several of the differences at birth were not clinically importantly different.

Line 230 to 232 should be stated in the past tense since these results should not be generalized beyond the study, particularly since it is an observational study and one with a very large loss to follow up.

Line 33 and the abstract: “SGA infants … were excluded.” but SGA infants are included in the Results in line 244.

Although widely used, it is important to mention that the “change in WAZ greater than or equal to +0.67 SD” has not been validated.

Reviewer #2: McLaughlin et al present an interesting study on infants’ growth velocity according to their birthweight. I have the following comments, questions and suggestion:

Abstract:

The abstract does not state clearly the aims of the study and the study design, that are instead specified at the end of the Introduction: line 99 “In this follow-up to the FLAG study we aimed to: i) determine whether AGA infants with slowing antenatal growth demonstrate catch-up-growth; ii) compare AGA catch-up growth to that of the SGA infants; and iii) determine in which postnatal time period catch up-growth occurs.” The abstract does not specify the statistic methods utilized to reach the study goals.

Methods

Line 130: what is the rationale behind using a different cut off for growth velocity than the one used in the FLAG study ?

Line 144: “the follow-up cohort was again divided into three groups: the SGA; the AGA with normal antenatal growth velocity (AGA-NG); and the AGA with low antenatal growth velocity (AGA-FGR)” This distinction into 3 study groups does not clearly appear from the abstract

Line 161: “To standardise growth velocity for the cohort, we divided the change in weight centile by the actual number of days between examinations and then multiplied by the exact number of days in that time epoch”. Could the authors clarify this part ?

Results

If the authors state in line 144 state that they are interested in studying 3 groups, why do they only present data on 2 groups in table 2 ?

Line 223 “We first analysed the relationship between antenatal growth velocity and postnatal growth velocity in AGA infants (Table 3). At birth there was a significant difference in weight

(p=0.0002), length (p=0.01) and BMI (p=0.0002) centiles between the AGA-FGR and AGA-NG infants.” Why is this important ? Isn’t such difference what discriminates between AGA-FGR and AGA-NG?

Table 3: again, why do the authors compare AGA-FGR and AGA-NG, instead of comparing the 3 groups AGA-FGR, AGA-NG, and SGA?

As postnatal growth is assessed as a series of measurements on the same subject over time, the authors may consider statistical tools such as linear mixed models for longitudinal data instead of comparing differences between 2 time points at a time

Table 4 considers 3 different study groups, why does table 3 consider only 2 ?

Table 5: as the authors consider 3 groups, they should use a statistica test that investigates differences in multiple groups and than specify further sub-group differences using a post hoc test.

Discussion: well written but reflects the weaknesses of the methodological design

Personally, I would consider the 3 study groups based on birthweight (AGA-NG, AGA-FGR and SGA) and compare their postnatal growth with a linear mixed model. The model provides a coefficient that summarizes the change in weight percentile between different time points. The model would allow to compare the growth speed of AGA-NG, AGA-FGR and SGA infants and it would also enable to study subgroup differences. Furthermore, the model would also allow to control for confounding

6. PLOS authors have the option to publish the peer review history of their article (what does this mean?). If published, this will include your full peer review and any attached files.

Reviewer #1: No

Reviewer #2: No

---

## [Author Response · Author response to Decision Letter 0]

12 May 2020

Dear Editors of PLOS One,

Re: PONE-D-19-31762

McLaughlin et al, ‘Appropriate-for-gestational-age infants who exhibit reduced antenatal growth velocity display postnatal catch up growth’

We thank the reviewers for their feedback and suggestions, which have improved the manuscript. Major statistical revisions have been performed for this resubmission as requested. Here we are pleased to provide our point-by-point responses, and submit this revised manuscript for consideration. 

For ease of review, we submit a manuscript with tracked changes, and also submit an identical clean copy. In this response letter, all page numbers and line numbers refer to those of the tracked changes copy.

We look forward to your further correspondence,

Dr Emma McLaughlin

Corresponding author

Reviewer #1

1. Abstract: While the Flag study used a more informed measure of EFW (used the slowing of growth) however this is not identified in the abstract and paper introduction and the abstract introduction only refers to weight <10th percentile. 

We have now added detail to the definition of “slowing growth” in the Background section of the abstract. Please see page 2, lines 22-24 which now read:

“Infants who demonstrate slowing growth during pregnancy are those that, cross estimated fetal weight centiles at serial ultrasound examinations.”

Detail regarding the definition of slowing growth is also given in the Methods section of the abstract. Page 2, lines 40-42 state:

“Predicted mean age-adjusted z-scores were then compared between three groups; SGA, AGA with low antenatal growth (AGA-FGR, loss of >20 customised estimated fetal weight centiles)”

2. The abstract needs to mention the high loss to follow which was ~ 50%, and the high proportion of GDM in those lost to follow up compared to respondents.

As requested, we have now included this information in the abstract, results section. We write, page 3, line 48:

“Of 158 (46%) infant growth records received,…” – hence reporting the follow up rate.

And on page 3, lines 49-50 we have added 

“Rates of gestational diabetes and SGA birthweight were higher in those lost to follow-up”. 

3. The word “responses” was used in the abstract but it is not clear what it means.

In response to this reviewer comment, we have now adjusted “responses” to “infant growth records” for greater clarity. (Abstract, Results subsection, page 3, line 48)

4. The abstract needs to mention that the postnatal anthropometric measurements were obtained by parent report.

The abstract has been altered to mention that the parents were asked to send the measurements, and also to clarify that the measurements themselves were taken by health professionals, not the parents themselves. It now reads, (Page 2, lines 33-35):

“To assess postnatal growth, we asked parents to send their infant’s growth measurements, up to two years post-birth, which are routinely collected through the state-wide Maternal-Child Health service.” 

5. The study makes the point that not all infants with a birthweight >10th percentile have avoided growth restriction. However, the counter problems exists as well, since the 10th percentile is an arbitrary statistical cut-off. Not all infants that are born SGA with a birthweight <10th percentile have utero placental insufficiency. The paper’s introduction should mention that some infants in the SGA category are small normal infants.

To meet this request of the reviewer we have now added a line to the introduction. Please see page 4, lines 70-75

“SGA… is the most commonly used proxy for FGR, as although not all SGA fetuses are growth restricted (some are constitutionally small), it does represent an at-risk cohort.

Being SGA is the greatest risk factor for stillbirth (3) but its legacy also extends into postnatal life among survivors.”

6. In contrast to the statement “strong evidence that the development of cardiometabolic disease in adulthood is related to early key growth periods”, some of these presumed effects may be seen due to the analysis methods used. It would be valuable to consider possible over adjustment as described in the following references:

1. Kramer MS, Zhang X, Dahhou M, Yang S, Martin RM, Oken E, et al. Does fetal growth restriction cause later obesity? Pitfalls in analyzing causal mediators as confounders. Am J Epidemiol. 2017 Apr;185(7):585–90.

2. Huxley R, Neil A, Collins R. Unravelling the fetal origins hypothesis: is there really an inverse association between birthweight and subsequent blood pressure? Lancet 2002 Aug;360(9334):659–65.

3. Paneth N, Ahmed F, Stein AD. Early nutritional origins of hypertension: a hypothesis still lacking support. J Hypertens Suppl. 1996 Dec;14(5):S121–9.

4. Tu Y-K, West R, Ellison GTH, Gilthorpe MS. Why evidence for the fetal origins of adult disease might be a statistical artifact: the “reversal paradox” for the relation between birth weight and blood pressure in later life. Am J Epidemiol. 2005;161:27–32.

While this study is based on belief in the fetal origins of adult health and disease theory, the reviewer raises a valid point in that this theory is not universally accepted. As such we have:

- Removed the word “strong” from the statement quoted by the reviewer above (page 4, line 82)

- Added a section to the strengths and limitations section of our discussion which states (Page 18, lines 369-374):

“Finally, this study interrogating fetal and postnatal growth rates is based upon the developmental origins of health and disease hypothesis. While this is a theory to which we ascribe, it is not universally accepted, with some research groups considering that the relationships between fetal growth and adult disease seen may be due to over-adjustment due to analysis method used (34-37)” 

7. It should be noted that there is a poor association between estimated foetal weight and birthweight.

In the limitations section of the Discussion we have now noted this by saying (Page 18, lines 374-377):

“In addition, the limitations of the original FLAG study have already been discussed (15), but relevant here is that estimated fetal weight is only accurate to within 10% of the true weight 80% of the time (38).”

8. This study used percentiles to evaluate antenatal growth, but percentiles become meaningless at the extremes. SD scores, which were used in some analyses, are better than percentiles as they are meaningful over the full range of the values.

In light of the comments from both Reviewer #1 and Reviewer #2, we have now repeated all of the analyses with the aid of a biostatistician Dr Richard Hiscock (MBiostat). All of the analyses have now been performed in using age-adjusted standard deviations z-scores rather than percentiles. You will see that the Methods and Results sections of the manuscript have both been largely edited.

9. What precision was used to calculate the standard deviation scores, was it the WHO monthly data or daily data?

We calculated infant age accurate to the number of days. When calculating z-scores we used age in months, accurate to precision of one decimal place – roughly equivalent to 3 days.

10. Analysis: Were any variables adjusted for in the regression analysis? It is not necessary to adjust, in fact it is appropriate to report crude analyses, it just needs to be clear if adjustments were made in any of the analyses

No adjustments were made. As such, we have clarified this in the methods, page 9, lines 206-207 where we now state:

“The relationships between EFW growth velocity and postnatal weight, length and BMI were assessed in two ways, without adjustment for any other variables:”

In addition, on page 10, lines 221-222 state:

“No adjustment for multiple comparisons for either significance testing or confidence interval width was performed.”

11. Line 188: “Fishers exact test was used was used to ascertain the relative risk of catch up growth” does not make sense since it is a statistical test.

This line (now Page 10, lines 214-222) has now been altered (it also reflects our updated statistical analyses performed) to read:

“Secondly, we assessed rates and relative risks of catch-up growth between the three groups. Catch-up growth was defined as an increase in weight age-adjusted z-score of �0.67 between birth and 12 months of age (29). Analysis used Fisher’s exact test across the three antenatal growth categories. If the null hypothesis of no overall difference in proportions was rejected, between pair testing was performed.”

12. Results: While there were significant differences in birth size between the FGR & NG infants, several of the differences at birth were not clinically importantly different.

In contrast to the reviewer, we take the view that a difference in mean birthweight of almost 20 centiles is clinically significant, in keeping with the point that those that slow in growth are growth restricted compared to their counterparts who maintain their EFW centile in utero. Nevertheless, Results, page 13, Lines 264-266 has been changed to specify ‘statistically significant’ when describing the differences seen, to maintain neutrality when presenting these results.

13. Line 230 to 232 should be stated in the past tense since these results should not be generalized beyond the study, particularly since it is an observational study and one with a very large loss to follow up.

This has been corrected to the past tense (now line 267-270, page 13).

14. Line 33 and the abstract: “SGA infants … were excluded.” but SGA infants are included in the Results in line 244.

Thank you for noticing this error. In the original FLAG study SGA infants were excluded from analysis, but they were included in the FLAG follow-up submitted here. As such, line 33 (Now line 35-36) has been corrected to only ‘infants with medical conditions’. 

15. Although widely used, it is important to mention that the “change in WAZ greater than or equal to +0.67 SD” has not been validated.

This is a valid point. A sentence has now been added to our methods section, (Page 8, lines 182 to 184) to reflect this, where we state:

“It is important to note however, that although widely used (30, 31), this definition of catch-up growth has not been formally validated.” 

 

Reviewer #2:

McLaughlin et al present an interesting study on infants’ growth velocity according to their birthweight. I have the following comments, questions and suggestion:

1. Abstract: The abstract does not state clearly the aims of the study and the study design, that are instead specified at the end of the Introduction: line 99 “In this follow-up to the FLAG study we aimed to: i) determine whether AGA infants with slowing antenatal growth demonstrate catch-up-growth; ii) compare AGA catch-up growth to that of the SGA infants; and iii) determine in which postnatal time period catch up-growth occurs.”

We have now adjusted the Background section of the Abstract to state more clearly the aims of the study as suggested by the reviewers. Page 2, lines 22-28 now read:

“Infants who demonstrate slowing growth during pregnancy are those that cross estimated fetal weight centiles at serial ultrasound examinations. These infants that slow in growth but are born appropriate-for-gestational age (AGA; >10th centile), exhibit antenatal, intrapartum and postnatal indicators of UPI. Here, we examine if and when these infants (labelled as AGA-FGR) also demonstrate catch-up growth like SGA infants, when compared with AGA infants with normal antenatal growth velocity (AGA-NG).”

2. The abstract does not specify the statistic methods utilized to reach the study goals.

We have now detailed the statistic tests utilised in the abstract (Pages 2-3, lines 36-45) where we state:

“From the measurements obtained we calculated age-adjusted z-scores for postnatal weight, length and body mass index (BMI; weight(kg)/height(m2)) at birth and 4, 8, 12, 18 and 24 months. We used linear spline regression modelling to predict mean weight, length, and BMI z-scores at intervals post birth. Predicted mean age-adjusted z-scores were then compared between three groups; SGA, AGA with low antenatal growth (AGA-FGR; loss of >20 customised estimated fetal weight centiles) and AGA-NG to determine if catch-up growth occurred. In addition we compared the rates of catch-up growth (defined as an increase in weight age-adjusted z-score of �0.67 over 1 year) between the groups with Fisher’s exact tests.”

3. Methods: Line 130: what is the rationale behind using a different cut off for growth velocity than the one used in the FLAG study ?

We used a cut-off of >20 centiles EFW loss instead of >30 centiles loss to ensure enough cases for a reasonable statistical comparison, given that we expected loss to follow up. We have now added more detail to this point in the Methods section, page 7, lines 143-152 which reads:

“The current study focussed on comparing antenatal and postnatal growth trajectory, not predictive performance for adverse outcome for which the original study was powered. Given the different outcome being interrogated, and expected loss to follow up, we used a more sensitive cut-off for growth velocity than the original FLAG study to ensure a sufficient number of cases of low antenatal growth velocity. Fetuses who exhibited an antenatal decrease in EFW of >20 customised centiles over eight weeks were defined as experiencing low antenatal growth velocity and to potentially be growth restricted (labelled as AGA-FGR). The remainder of the AGA fetuses were defined as demonstrating normal antenatal growth (AGA-NG)."

4. Line 144: “the follow-up cohort was again divided into three groups: the SGA; the AGA with normal antenatal growth velocity (AGA-NG); and the AGA with low antenatal growth velocity (AGA-FGR)” This distinction into 3 study groups does not clearly appear from the abstract

We have now adjusted the abstract background and methods to clarify the distinction into the three study groups. 

The Background section of the abstract, page 2, lines 26-28 now states:

“Here, we examine if and when these infants (labelled as AGA-FGR) also demonstrate catch-up growth like SGA infants, when compared with AGA infants with normal antenatal growth velocity (AGA-NG).

The Methods section of the abstract, pages 2-3, lines 40-43 now state:

“Predicted mean age-adjusted z-scores were then compared between three groups; SGA, AGA with low antenatal growth (AGA-FGR, loss of >20 customised estimated fetal weight centiles), and AGA-NG to determine if catch-up growth occurred.

5. Line 161: “To standardise growth velocity for the cohort, we divided the change in weight centile by the actual number of days between examinations and then multiplied by the exact number of days in that time epoch”. Could the authors clarify this part ?

This sentence has been removed after inclusion of new statistical methods.

6. Line 223 “We first analysed the relationship between antenatal growth velocity and postnatal growth velocity in AGA infants (Table 3). At birth there was a significant difference in weight (p=0.0002), length (p=0.01) and BMI (p=0.0002) centiles between the AGA-FGR and AGA-NG infants.” Why is this important? Isn’t such difference what discriminates between AGA-FGR and AGA-NG?

AGA-FGR and AGA-NG are defined by their antenatal growth velocity and that they were born at birthweight >10th centile, the groups are not defined by having different birthweight or other anthropometric measurements at birth. AGA-FGR infants, exhibited an antenatal decrease in EFW of greater than 20 customised centiles between 28- and 36-weeks’ gestation, while AGA-NG infants did not. Despite the difference in antenatal growth rate, both groups were born with weight greater than 10th centile (AGA). Therefore, if we had not tracked antenatal growth velocity, they would be considered simply equal AGA infants. These initial differences in birth measurements are therefore important to note, as even by themselves they add weight to the evidence that AGA infants who have demonstrated slowing growth in utero are growth restricted/subject to uteroplacental insufficiency relative to their AGA counterparts who maintain their EFW centile throughout gestation. This was also seen in the initial FLAG study, but we found it prudent to report this again as evidence maintained even here among a different, smaller sub-cohort. In addition, that the infants of the AGA-FGR group are smaller at birth, and then no longer significantly smaller at 4 months of age is evidence of catch-up growth. 

Results:

7. If the authors state in line 144 state that they are interested in studying 3 groups, why do they only present data on 2 groups in table 2 ?

SGA fetuses and infants are already known to be a cohort at increased risk. In contrast, AGA fetuses and infants who have demonstrated slowing antenatal growth are the cohort which in this study we primarily are investigating to see if they demonstrate catch-up growth, as further evidence of uteroplacental insufficiency compared to AGA fetuses who maintain their EFW centile across gestation. As such, we compared the maternal characteristics between AGA-FGR and AGA-NG groups. 

This is now clarified in the Results section, page 11, lines 238-241 where we now state:

“Given our primary question concerned whether AGA-FGR infants demonstrate catch-up growth compared to the AGA-NG, as a sign of UPI occurring among the AGA, we compared their maternal and pregnancy characteristics (Table 1).”

8. Table 3: again, why do the authors compare AGA-FGR and AGA-NG, instead of comparing the 3 groups AGA-FGR, AGA-NG, and SGA? As postnatal growth is assessed as a series of measurements on the same subject over time, the authors may consider statistical tools such as linear mixed models for longitudinal data instead of comparing differences between 2 time points at a time.

In light of the reviewer’s comments, we have now reanalysed the data with different statistical methodology, see below.

9. Table 4 considers 3 different study groups, why does table 3 consider only 2? Table 5: as the authors consider 3 groups, they should use a statistical test that investigates differences in multiple groups and then specify further sub-group differences using a post hoc test. 

After our reanalysis, we now present 2 tables of catch-up growth related data. Both include all three antenatal growth groups. 

Table 3 (previously Table 5) still presents the results of between pair testing, but the results of testing the three groups together are presented in the text when we state (Results, page 16, lines 316-321):

“When catch-up growth was assessed as a dichotomous outcome (change in WAZ �+0.67) between birth and 12 months of age between all three groups, a significant difference was found (p=0.004). When between pair testing was then performed, the SGA and AGA-FGR cohorts were both found to be at increased risk of catch-up growth when compared to the AGA-NG group (Table 3). 

10. Personally, I would consider the 3 study groups based on birthweight (AGA-NG, AGA-FGR and SGA) and compare their postnatal growth with a linear mixed model. The model provides a coefficient that summarizes the change in weight percentile between different time points. The model would allow to compare the growth speed of AGA-NG, AGA-FGR and SGA infants and it would also enable to study subgroup differences. Furthermore, the model would also allow to control for confounding

In response to Reviewer #2’s points 8-10 we have enlisted the expertise of biostatistician Dr Richard Hiscock (MBiostat) and we have reanalysed the data. You will note that the methods and the results sections of the manuscript have been rewritten to reflect these reanalyses.

Specifically, the Statistical Analysis section of the Methods now outlines the new analyses performed, reading (pages 9-10, lines 195-222):

“Antenatal growth velocity and catch-up growth

Growth curve modelling was performed using linear spline regression, to fit curves plotting the change in predicted mean weight, length and BMI z-scores over time, for each antenatal growth category. User determined knots were placed at five time points (4, 6, 12, 18 and 24 months). We also fitted growth curves using both: restricted cubic spline model with knots placed at the quartiles of the time distribution and an interaction between time and antenatal growth category; and locally weighted kernel regression. All statistical methods produced extremely similar results, reinforcing the validity of our linear spline regression approach (Data not shown, available upon request). 

The relationships between EFW growth velocity and postnatal weight, length and BMI were assessed in two ways, without adjustment for any other variables. First predicted mean difference and associated 95% confidence limits for weight, height and BMI z-scores were calculated at birth, 2, 4, 6, 12, 18 and 24 months for the three comparisons between antenatal growth categories (SGA & AGA-FGR, SGA & AGA-NG, and AGA-FGR & AGA-NG). We defined catch-up growth as occurring if the SGA or AGA-FGR group infants were significantly smaller than the AGA-NG group infants in weight, length and/or BMI at one time point, followed by no significant difference in infant size months later. Secondly, we assessed rates and relative risks of catch-up growth between the three groups. Catch-up growth was defined as an increase in weight age-adjusted z-score of �0.67 between birth and 12 months of age (29). Analysis used Fisher’s exact test across the three antenatal growth categories. If the null hypothesis of no overall difference in proportions was rejected, between pair testing was performed. Statistical analyses were performed using GraphPad Prism software version 7.0d for Mac OS X (32) and the nonparametric series regression suite within Stata v16 (33). Significance level was two-sided and set at 0.05. No adjustment for multiple comparisons for either significance testing or confidence interval width was performed.”

And the results presented reflect this analysis. We now include two new figures, Figures 1 and 2, which show the raw infant measurement data as well as the predicted means (and 95% confidence intervals) for infant weight, length and BMI for each of the three antenatal growth groups together. 

11. Discussion: Well written but reflects the weaknesses of the methodological design.

We thank the reviewer most sincerely for these suggestions. In this revision we have tried to address the major weaknesses in design. In response to your suggestion we have conducted considerable further statistical analyses to compare all three groups. The additional statistical analyses performed are all outlined in the methods and results sections of the manuscript which are now considerably different to that of the manuscript originally submitted.

---

## [Decision Letter · Decision Letter 1]

11 Jun 2020

PONE-D-19-31762R1

Appropriate-for-gestational-age infants who exhibit reduced antenatal growth velocity display postnatal catch-up growth

PLOS ONE

Dear Dr. McLaughlin,

Thank you for submitting your manuscript to PLOS ONE, and having addressed most of the remarks made by the reviewers and the editor in your revised manuscript. After careful consideration, the manuscript does not fully meet PLOS ONE’s publication criteria as it currently stands, despite its revision. Therefore, we invite you to submit a revised version of the manuscript that addresses the points raised during the review process.

We ask you to take into consideration the last remarks made by Reviewer No 1, and submit a second revision of your manuscript.

We look forward to receiving your revised manuscript.

Kind regards,

Umberto Simeoni

Academic Editor

PLOS ONE

Reviewers' comments:

Reviewer's Responses to Questions

**Comments to the Author**

1. If the authors have adequately addressed your comments raised in a previous round of review and you feel that this manuscript is now acceptable for publication, you may indicate that here to bypass the “Comments to the Author” section, enter your conflict of interest statement in the “Confidential to Editor” section, and submit your "Accept" recommendation.

Reviewer #1: (No Response)

2. Is the manuscript technically sound, and do the data support the conclusions?

Reviewer #1: Yes

3. Has the statistical analysis been performed appropriately and rigorously? 

Reviewer #1: Yes

4. Have the authors made all data underlying the findings in their manuscript fully available?

Reviewer #1: Yes

5. Is the manuscript presented in an intelligible fashion and written in standard English?

Reviewer #1: Yes

6. Review Comments to the Author

Reviewer #1: This study is generally well written. This study reinforces the point that the common cut off for SGA births as <10th percentile is an arbitrary cut off that has limitations and observed that infants who were diagnosed with growth restriction during pregnancy grow at faster rates in their first 4 months of life. The biggest concern with this manuscript is the acceptance of the Barker hypothesis, which is repeated numerous times throughout the manuscript. I encourage these authors to consider a critical appraisal of that hypothesis.

EFW growth velocity defined as a fall of 20 and 30 centiles in line 112 and 133. Please clarify.

Paragraph beginning on Line 80: The ‘Barker hypothesis’ (13) proposes that the relationship between FGR and adult disease is due to fetal programming in utero. � there is good evidence to show that the ‘Barker hypothesis’ is not supported and due to over-adjustment in regression analysis (1-4), which is known to distort regression analysis results (5,6) so this section should be revised to mention that: the ‘Barker hypothesis’ is only a hypothesis, several mechanisms have been proposed for how slow followed by rapid growth might lead to adult medical issues, and how based on this hypothesis and evidence from studies that used the Barker regression methods suggest that catch-up growth could be part of the mechanism.

1. Kramer MS, Zhang X, Dahhou M, Yang S, Martin RM, Oken E, et al. Does fetal growth restriction cause later obesity? Pitfalls in analyzing causal mediators as confounders. Am J Epidemiol. 2017 Apr;185(7):585–90.

2. Huxley R, Neil A, Collins R. Unravelling the fetal origins hypothesis: is there really an inverse association between birthweight and subsequent blood pressure? Lancet 2002 Aug;360(9334):659–65.

3. Paneth N, Ahmed F, Stein AD. Early nutritional origins of hypertension: a hypothesis still lacking support. J Hypertens Suppl. 1996 Dec;14(5):S121–9.

4. Tu Y-K, West R, Ellison GTH, Gilthorpe MS. Why evidence for the fetal origins of adult disease might be a statistical artifact: the “reversal paradox” for the relation between birth weight and blood pressure in later life. Am J Epidemiol. 2005;161:27–32.

5. Schisterman EF, Cole SR, Platt RW. Overadjustment bias and unnecessary adjustment in epidemiologic studies. Epidemiology [Internet]. 2009;20(4):488–95.

6. Ananth C V., Schisterman EF. Confounding, causality, and confusion: the role of intermediate variables in interpreting observational studies in obstetrics. Am J Obstet Gynecol [Internet]. 2017;217(2):167–75.

Catch-up growth is referred to negatively in lines 264, 282, 343, 376-7 and 393-4, which should be revised to remove the value judgements, especially given the large percentile anthropometric recoveries seen in this study (produced a distribution of sizes that were similar to the non-growth restricted infants) which suggests that the infants tend to grow in this manner. This negative messaging about catch-up growth would not be desirable to be presented to parents to have parents restrict infants’ feedings to prevent this catch up growth, as further suboptimal growth would not likely support good brain growth and it could harm the relationship between the parent(s) and the child.

Line 264: the SGA and AGA-FGR cohorts were both found “to be at increased risk” of catch-up-growth when compared to the AGA-NG group � since catch up growth is not necessarily undesirable, I suggest this be changed to “have a higher probability”. It would be desirable also to refer to the RR as the risk ratio.

Lines 357-364 – while rapid weight gain has been associated with later obesity, these studies should be examined for whether or not they adjusted for the social determinants of health, since many social determinants can put people at risk of living in an obesogenic environment, and may be the cause of both early weight gain and later obesity.

This paper would be enhanced with the median measurements for all 3 groups for wt, L and HC plotted on the WHO growth charts. An App such as this could be used https://apps.cpeg-gcep.net/growth02/

The results could appear considerably differently if change in SD-scores were were used instead of centiles, since SD-scores are linear across the normal curve distribution while centiles increment in a non-linear fashion. For example, between 0, 1, 2 and 3 SD-scores, there are 34%, 13%, 3% and <0.1% of the normal curve distribution.

Minor points

“in the face of” = colloquial, not plain language

178 recruitment bias: it would be preferable to use the words selection bias in the language used by the Cochrane Collaboration

Table 1 – what are the units for EFW change in 8 weeks?

It is good practice to use 2 digits in the p-values on the right side of the decimal point for NS findings and 3 digits for significant findings.

Line 227: (-1.9 vs. -13.8, p=0.04): interesting – looks like the NG group had regression to the mean or some catch down growth

Line 230 – missing a “p=”?

Line 232 – should be past tense: “occurs”,

Line 351 – should be past tense

How did the nurses assign the percentiles? Was it using growth charts or using a computer tool? Which was used will help to describe the precision or lack thereof.

7. PLOS authors have the option to publish the peer review history of their article (what does this mean?). If published, this will include your full peer review and any attached files.

Reviewer #1: No

---

## [Author Response · Author response to Decision Letter 1]

9 Aug 2020

Dear Editors of PLOS One,

Re: PONE-D-19-31762

McLaughlin et al, ‘Appropriate-for-gestational-age infants who exhibit reduced antenatal growth velocity display postnatal catch up growth’

We thank the reviewer for their feedback and suggestions.

The line numbers, and much of the content, referenced by the reviewer belong to the original manuscript (submitted on 14th November 2019), not to the subsequent revision, submitted on 12th May 2020. Many of the points raised had thus already been addressed in this revision. We wonder whether it is possible that the reviewer received- or reviewed- the original submission in error rather than the revised version with all of our accompanying responses. Nevertheless, here we are pleased to again provide point-by-point responses to the feedback and submit this second revision of our manuscript for consideration.

For ease of review, we submit a manuscript with tracked changes, and also submit an identical clean copy. Where relevant, we have attempted to provide both the line number from the original manuscript, and the line numbers from (the tracked changes copy of) the previously submitted revision. We apologise that this is somewhat confusing, but it was the only way to address the comments of the reviewer which matched the line numbers of the original (not revised) version. We hope that the changes summarised here meet with the approval of the reviewers. 

We look forward to your further correspondence,

Dr Emma McLaughlin

Corresponding author.

 

1. This study reinforces the point that the common cut off for SGA births as <10th percentile is an arbitrary cut off that has limitations and observed that infants who were diagnosed with growth restriction during pregnancy grow at faster rates in their first 4 months of life

To meet this request of the reviewer we previously added a line to the introduction in our first revision. This point is discussed where we state (page 4, lines 68-74): 

“UPI results in failing growth, commonly manifesting as low birthweight centile. As such, small-for-gestational-age (SGA; estimated fetal weight (EFW), or birthweight <10th centile) is the most commonly used proxy for FGR, as although not all SGA fetuses are growth restricted (some are constitutionally small), it does represent an at-risk cohort. Being SGA is the greatest risk factor for stillbirth (3) …”

2. The biggest concern with this manuscript is the acceptance of the Barker hypothesis, which is repeated numerous times throughout the manuscript. I encourage these authors to consider a critical appraisal of that hypothesis. Paragraph beginning on Line 80: The ‘Barker hypothesis’ (13) proposes that the relationship between FGR and adult disease is due to fetal programming in utero. There is good evidence to show that the ‘Barker hypothesis’ is not supported and due to over-adjustment in regression analysis (1-4), which is known to distort regression analysis results (5,6) so this section should be revised to mention that: the ‘Barker hypothesis’ is only a hypothesis, several mechanisms have been proposed for how slow followed by rapid growth might lead to adult medical issues, and how based on this hypothesis and evidence from studies that used the Barker regression methods suggest that catch-up growth could be part of the mechanism.

1. Kramer MS, Zhang X, Dahhou M, Yang S, Martin RM, Oken E, et al. Does fetal growth restriction cause later obesity? Pitfalls in analysing causal mediators as confounders. Am J Epidemiol. 2017 Apr;185(7):585–90.

2. Huxley R, Neil A, Collins R. Unravelling the fetal origins hypothesis: is there really an inverse association between birthweight and subsequent blood pressure? Lancet 2002 Aug;360(9334):659–65.

3. Paneth N, Ahmed F, Stein AD. Early nutritional origins of hypertension: a hypothesis still lacking support. J Hypertens Suppl. 1996 Dec;14(5):S121–9.

4. Tu Y-K, West R, Ellison GTH, Gilthorpe MS. Why evidence for the fetal origins of adult disease might be a statistical artifact: the “reversal paradox” for the relation between birth weight and blood pressure in later life. Am J Epidemiol. 2005;161:27–32.

5. Schisterman EF, Cole SR, Platt RW. Overadjustment bias and unnecessary adjustment in epidemiologic studies. Epidemiology [Internet]. 2009;20(4):488–95.

6. Ananth C V., Schisterman EF. Confounding, causality, and confusion: the role of intermediate variables in interpreting observational studies in obstetrics. Am J Obstet Gynecol [Internet]. 2017;217(2):167–75.’

The reviewer raised this point in their review of the original manuscript, and we revised the manuscript accordingly. In light of these comments we wonder if the reviewer may have not received, or reviewed, the revised version of the manuscript.

While this study is based on belief in the fetal origins of adult health and disease theory (line 80 in the original manuscript, not in the revised version of the manuscript (this theory is now discussed from line 87 in the revised manuscript), this theory is not universally accepted. We believe we addressed this point in our first revision where we added a section to the strengths and limitations section of our discussion utilising the references suggested by the reviewer at time of their first review, which states (Page 19, lines 378-383):

“Finally, this study interrogating fetal and postnatal growth rates is based upon the developmental origins of health and disease hypothesis. While this is a theory to which we ascribe, it is not universally accepted, with some research groups considering that the relationships between fetal growth and adult disease seen may be due to over-adjustment due to analysis method used (35-38)” 

We have not made any further changes, as we believe the reviewer may not have had the benefit of seeing the revisions made already.

3. EFW growth velocity defined as a fall of 20 and 30 centiles in line 112 and 133. Please clarify.

Again, the use of differing definitions of EFW growth velocity; loss of >30 centiles in the original FLAG study and loss of >20 centiles in this FLAG follow-up study (lines 112 and 133 in the original manuscript and line 119 and 149 in the revised manuscript, respectively) was addressed in our first revision. We used a broader cut-off of >20 centiles EFW loss instead of >30 centiles loss to ensure enough cases for reasonable statistical comparison, given that we expected loss to follow up. At the time of our first revision we added more detail regarding our use of the different cut-offs in the Methods section, page 7, lines 144-153 which reads:

“The current study focussed on comparing antenatal and postnatal growth trajectory, not predictive performance for adverse outcome for which the original study was powered. Given the different outcome being interrogated, and expected loss to follow up, we used a more sensitive cut-off for growth velocity than the original FLAG study to ensure a sufficient number of cases of low antenatal growth velocity. Fetuses who exhibited an antenatal decrease in EFW of >20 customised centiles over eight weeks were defined as experiencing low antenatal growth velocity and to potentially be growth restricted (labelled as AGA-FGR). The remainder of the AGA fetuses were defined as demonstrating normal antenatal growth (AGA-NG)."

We have therefore not made any further changes, regarding this point.

4. Catch-up growth is referred to negatively in lines 264, 282, 343, 376-7 and 393-4, which should be revised to remove the value judgements, especially given the large percentile anthropometric recoveries seen in this study (produced a distribution of sizes that were similar to the non-growth restricted infants) which suggests that the infants tend to grow in this manner. This negative messaging about catch-up growth would not be desirable to be presented to parents to have parents restrict infants’ feedings to prevent this catch up growth, as further suboptimal growth would not likely support good brain growth and it could harm the relationship between the parent(s) and the child. E.g. Line 264: the SGA and AGA-FGR cohorts were both found “to be at increased risk” of catch-up-growth when compared to the AGA-NG group - since catch up growth is not necessarily undesirable, I suggest this be changed to “have a higher probability”. 

We agree that catch-up-growth in not necessarily undesirable. As recognised in the introduction, it is thought to be beneficial for neurodevelopment (Page 4, lines 81-82). To meet the request of the reviewer regarding the language in lines 264, 282, 343, 376-7 and 393-4 in the original manuscript (lines 320, 338, 409-410, 446-447 and 461-463 in the revised manuscript, respectively) we have adjusted the wording as the reviewer suggested in these sentences so that they now read with a more neutral tone, as follows:

Page 16, lines 322-325:

“When between pair testing was then performed, the SGA and AGA-FGR cohorts were both found to have a higher probability of catch-up growth when compared to the AGA-NG group (Table 3). For the SGA cohort the probability was approximately tripled, and for the AGA-FGR cohort, it was doubled.”

Page 18, lines 343-344:

“We have demonstrated that catch-up growth – hypothesised to be associated with some long-term health decrements”

Page 21, lines 419-421:

“While their findings support our results, we know that both SGA and preterm AGA infants are more likely to demonstrate catch-up growth (22, 43)”

We elected not to change the language used in the final two sentences highlighted by the reviewer, as these speak directly to negative health outcomes, and therefore the word “risk” is appropriate in this context. We highlight however that these sentences both speak hypothetically using the word “may”. They read as follows (page and line numbers reference their location in this, the second revision):

Page 22, lines 461-463:

“These infants may therefore be at increased risk of the antenatal, intrapartum, postnatal, childhood and adulthood complications that FGR and catch-up growth confer compared to AGA-NG infants” 

And conclusion, page 24, lines 485-487

“Our findings suggest that these infants may also be at increased risk of developing the long-term health decrements traditionally associated with infants born SGA”

5. It would be desirable also to refer to the RR as the risk ratio.

We have chosen to continue to refer to the RR as ‘relative risk’ for continuity of language with our previous studies. 

6. Lines 357-364 – while rapid weight gain has been associated with later obesity, these studies should be examined for whether or not they adjusted for the social determinants of health, since many social determinants can put people at risk of living in an obesogenic environment, and may be the cause of both early weight gain and later obesity.

We acknowledge that there is an association between low socioeconomic status and low birth weight, as well as low socioeconomic status and obesity in developed countries. In lines 357-364 in the original manuscript (now corresponding to pages 21-22, lines 437-449), the referenced articles by Settler et al. (46) and Monteiro et al. (47) adjusted for socioeconomic status in their analyses. To make this point clear, lines 437-445 now read as: 

“Rotevatn et al. (45), Stettler et al. (46) and Monteiro et al. (47) found that rapid weight gain in the first year of life was associated with being overweight at 2, 7, and 14-16 years of age, respectively, independent of socioeconomic status. This is supported by a recent meta-analysis which found that the odds of being overweight from childhood to adulthood was 3.66 times greater in infants who experienced catch-up growth (48).”

7. This paper would be enhanced with the median measurements for all 3 groups for wt, L and HC plotted on the WHO growth charts. An App such as this could be used https://apps.cpeg-gcep.net/growth02/

The data analysis has largely been repeated after the first review of the original manuscript, and the reporting of our results along with it. Our figures now plot the predicted mean z-scores for weight, length and BMI for all three infant groups together – similar to the suggestion made by the reviewer here. We have elected not to plot graphs onto the WHO growth charts however as these plot centiles, and we now report z-scores only, and further, the app suggested for use does not allow us to plot the three groups together on one graph. Therefore, we have not altered the figure representation of our data from our first revision of the paper which were a major change at time of first review.

8. The results could appear considerably differently if change in SD-scores were used instead of centiles, since SD-scores are linear across the normal curve distribution while centiles increment in a non-linear fashion. For example, between 0, 1, 2 and 3 SD-scores, there are 34%, 13%, 3% and <0.1% of the normal curve distribution.

We already changed all of our reporting to SD scores (z-scores) at the time of the first revision. Given that only SD scores have been used in the manuscript since that time for assessment postnatal growth, we have not made any further changes to this point.

9. “in the face of” = colloquial, not plain language

The wording has been changed on page 4, lines 68-69 to read:

“Decreased nutrient and oxygen supply consequent to UPI results in failing growth, commonly manifesting as a low birthweight centile”

10. 178 recruitment bias: it would be preferable to use the words selection bias in the language used by the Cochrane Collaboration

The wording has been changed on page 9, lines 189-192 which now reads:

“Maternal characteristics and birth outcome data were compared between both: recruited follow-up study participants and the eligible women who did not participate, to check for selection bias; and cases of low antenatal velocity and the remainder of the AGA cohort, the subjects of our primary comparison”

Again, in our results we have changed the wording (page 11, lines 232-233) to read:

“We compared the maternal characteristics and delivery outcomes of the 156 women we recruited to the 189 non-responders to assess for selection bias (S1 Table).

11. Table S1: What are the units for EFW change in 8 weeks?

The units for EFW change in 8 weeks has been added to Table S1

12. Table S1: It is good practice to use 2 digits in the p-values on the right side of the decimal point for NS findings and 3 digits for significant findings.

In light of this point, the number of decimal points presented for p-values in Table S1 are now consistent.

13. Line 227: (-1.9 vs. -13.8, p=0.04): interesting – looks like the NG group had regression to the mean or some catch down growth

Yes, we agree with the reviewer on this point which is why this finding is described in the manuscript on pages 14-15, lines 298-302:

“…the shape of the growth curves in Figs 1 and 2 show that the AGA-FGR group demonstrated a similar, physiological fall in z-scores like the AGA-NG group, but of a lesser magnitude – leading to their catch-up growth. In contrast, the SGA cohort’s catch-up growth took the form of a steady increase in z-scores without the physiological fall seen among the AGA.”

14. Line 230 – missing a “p=”?

The sentence previously containing the missing “p=" (previously line 230 in the original manuscript, not the first revision) no longer exists in the current (revised) manuscript.

15. Line 232 – should be past tense: “occurs”,

Line 232 (now page 13, line 272) was corrected to past tense in the previous manuscript revision and therefore has required alteration in this revision

16. Line 351 – should be past tense

Line 351 in the original manuscript (lines 417-419 in the first revision of the manuscript) now has now been corrected to the past tense on page 21, lines 428-430:

“While catch-up growth was more evident in weight than length between birth and 4 months in our cohort, at no point in the first 2 years was BMI greater in AGA-FGR than in AGA-NG infants.”

17. How did the nurses assign the percentiles? Was it using growth charts or using a computer tool? Which was used will help to describe the precision or lack thereof

Maternal child health nurses did not assign percentiles. Raw measurements of length, weight and head circumference were taken in centimetres to 1 decimal place or to the nearest gram for weight. We obtained the records of these measurements and converted them to z-scores using the WHO child growth standards software as described on pages 6 and 8 in lines 126-131 and 168-170 (transcribed below), respectively. This software is similar to that recommended by the reviewer in point number 5 (https://www.who.int/childgrowth/software/en/).

“This collected… the infant weight, length and head circumference measurements. These measurements are routinely performed during maternal-child-health visits at 1, 2, 4 and 8 weeks; 4, 8, 12 and 18 months; and at 2 years of age and recorded in the child’s analogue ‘My Health, Learning & Development Record’…

Then “Using the WHO child growth standards software (29), we calculated standard deviation z-scores, adjusted for sex and age, for weight, length and body mass index (BMI; weight(kg)/height(m)2)”

---

## [Editor Report · Decision Letter 2]

24 Aug 2020

Appropriate-for-gestational-age infants who exhibit reduced antenatal growth velocity display postnatal catch-up growth

PONE-D-19-31762R2

Dear Dr. McLaughlin,

We’re pleased to inform you that your manuscript has been judged scientifically suitable for publication and will be formally accepted for publication once it meets all outstanding technical requirements.

Kind regards,

Umberto Simeoni

Academic Editor

PLOS ONE
---

## [Editor Report · Acceptance letter]

28 Aug 2020

PONE-D-19-31762R2 

Appropriate-for-gestational-age infants who exhibit reduced antenatal growth velocity display postnatal catch-up growth 

Dear Dr. McLaughlin:

I'm pleased to inform you that your manuscript has been deemed suitable for publication in PLOS ONE. Congratulations! Your manuscript is now with our production department. 

Kind regards, 

on behalf of

Dr. Umberto Simeoni 

Academic Editor

PLOS ONE